# Predicting seasonal influenza using supermarket retail records

**Ioanna Miliou**[1,2]*, **Xinyue Xiong**[3], **Salvatore Rinzivillo**[2], **Qian Zhang**[3], **Giulio Rossetti**[2], **Fosca Giannotti**[2], **Dino Pedreschi**[1], **Alessandro Vespignani**[3]

**1** University of Pisa, Pisa, Italy, **2** ISTI-CNR, Pisa, Italy, **3** Northeastern University, Boston, Massachusetts, United States of America

* ioanna.miliou@for.unipi.it

**Data Availability Statement:** The majority of our data and code is available at https://github.com/jeannetm/predict_influenza_with_retail_records. However, we do not include the COOP files regarding the retail records as they are not publicly

## Abstract

Increased availability of epidemiological data, novel digital data streams, and the rise of powerful machine learning approaches have generated a surge of research activity on real-time epidemic forecast systems. In this paper, we propose the use of a novel data source, namely retail market data to improve seasonal influenza forecasting. Specifically, we consider supermarket retail data as a proxy signal for influenza, through the identification of sentinel baskets, i.e., products bought together by a population of selected customers. We develop a nowcasting and forecasting framework that provides estimates for influenza incidence in Italy up to 4 weeks ahead. We make use of the Support Vector Regression (SVR) model to produce the predictions of seasonal flu incidence. Our predictions outperform both a baseline autoregressive model and a second baseline based on product purchases. The results show quantitatively the value of incorporating retail market data in forecasting models, acting as a proxy that can be used for the real-time analysis of epidemics.

## Author summary

Seasonal influenza is a major burden to the health care systems of countries. Machine learning approaches and data from external sources are increasingly used for flu forecasting in recent years. In this study, we explore whether the inclusion of retail records in a predictive model improves seasonal influenza forecasting. Specifically, we consider supermarket retail data as a proxy signal for influenza, through the identification of sentinel baskets, i.e., products bought together by a population of selected customers. We develop a nowcasting and forecasting framework that provides estimates for influenza incidence in Italy up to 4 weeks ahead. Our predictions outperform the baseline approaches thus proving the added value of incorporating retail market data in forecasting models.

## Introduction

Recent years have seen a growing interest in generating real-time epidemic forecasts through novel digital data streams and machine learning approaches. Seasonal influenza forecasting

available data. They are accessible though through the SoBigData Catalogue in this link: http://data. d4science.org/ctlg/ResourceCatalogue/retail_ market_data. SoBigData is the European Research Infrastructure for Big Data and Social Mining. For more details about the EU Project you can visit the Project Site: http://www.sobigdata.eu/ Due to privacy and confidentiality reasons the access is only on-site visit.

**Funding:** IM, SR, GR, FG, DP were partially supported by the European Community's H2020 Program, grant agreement # 654024 "SoBigData: Social Mining and Big Data Ecosystem" and grant agreement #871042 "SoBigData++: European Integrated Infrastructure for Social Mining and Big Data Analytics". IM was partially supported by a "Grant for Young Mobility" (GYM 2018) of ISTI-CNR. AV and XX were partially supported by the National Institute of General Medical Sciences of the National Institutes of Health under Award Number R01GM130668. The funders had no role in study design, data collection and analysis, decision to publish, or preparation of the manuscript.

**Competing interests:** The authors have declared that no competing interests exist.

approaches are leading the way in this rapidly advancing research landscape. Seasonal influenza is still a major burden to the health care systems of countries with 3 to 5 million infected, and 290,000–650,000 deaths caused by influenza worldwide every year [1]. For this reason, the US Centers for Disease Control and Prevention (CDC) formally pioneered infectious disease forecasting by starting the Flusight consortium focused on prediction of seasonal flu incidence. The CDC seasonal influenza challenge has been remarkably successful in maintaining momentum for a coordinated focus on the operational implementation of disease forecasting. Simultaneously, it fuels the research on developing forecasting models based both on traditional surveillance systems such as influenza-like illness (ILI) incidence captured by the network of outpatient clinics, and novel digital data streams such as search engine queries and social media [2–7]. In this context the use of machine learning techniques has received considerable attention [8], and although the use of novel digital data streams as proxy data for disease forecasting did show evident limitations in early approaches, the use of multiple data sources and ensemble of models is now defining the second generation of forecasting tools defining the state of the art in the field.

The pioneer in the use of machine learning and proxy data for flu forecasting has been the famous *Google Flu Trends (GFT)* platform. The platform was providing forecasts of the current level of influenza-like illness (ILI) incidence in the USA by using search engine queries associated with flu-related keywords [9]. The initial successes of the platform were followed by a number of problems and inaccuracies discussed in several in-depth analyses of the GFT results [10–12].

The failure of GFT was, however, success in disguise, as it stimulated the research community to develop novel ways to integrate proxy data that have considerably improved on the initial results. In particular, several research efforts were devoted to the exploration and combination of additional novel data streams—such as Twitter, hospital records, Wikipedia searches, anonymous influenza test or syndromic records, to name a few—in their predictive system of seasonal influenza [13–31]. Similar data streams have been explored for other epidemics like Dengue [32], Zika [33], hand-foot-mouth diseases [34], Ebola, plague, and yellow fever [35]. Most recently, forecasting and nowcasting models have been employed on different scales to address the COVID-19 pandemic crisis with a wide range of data proxies: the Internet search activity, news media, social media, social networking sites, wearable devices, etc. [36–41].

Along with the traditional flu surveillance system data, other innovative participatory surveillance systems, which aim at capturing influenza activity directly from the general population through Internet-based surveys, were developed and integrated into the forecasting approaches; *Influenzanet*, a network of Web platforms running in 11 European countries [42, 43], *FluNearYou* in the United States [44–46], and *FluTracking* in Australia [47–49]. These data are also used along with mobility and sociodemographic data to define new strategies for influenza incidence inference, such as mobility traces from mobile phones and the daily self-reported flu-like symptoms [50], or mobility data and the underlying social network [51].

Novel digital data streams and data collections approaches have also been used in the context of flu forecasting based on mechanistic models, defined as methods that include the mechanism of transmission of infection from an infected to an uninfected host. In these approaches, historical surveillance data, mobility, and socioeconomic data, along with novel digital data streams, are used to calibrate and initialize mechanistic models in a way akin to classic weather forecasting models [2, 5, 17, 52]. While these models provide access to the flu transmission mechanisms, they challenge us in the understanding of the assumptions and inputs employed in the definition of the transmission dynamics, and how these choices affect

the forecast results. The influenza challenge initiated by the US Centers for Disease Control and Prevention (CDC) in the 2013/2014 winter season has been a major initiative that fostered the research in infectious disease forecasting in a formal way and led to modeling advances that have been integrated into the CDC's operations [53]. Among the most relevant results that emerged from this coordinated effort, involving more than three dozen different forecasting methods [54, 55], is the evidence that ensemble forecasts that combine outputs from different models appear to offer the best trade-off between reliability and accuracy of the results [56–58].

Despite the advances in the field, more work is needed to rigorously understand the relationships among forecasting accuracy, modeling approaches, and data availability. Furthermore, most of the research has focused on a limited number of countries outside the USA, and there is a dire need for more systematic investigations of the feasibility and performance of flu forecasts across the world.

Here we propose a novel, high quality data source, particularly retail market data, as a proxy for seasonal influenza nowcasts and forecasts. The assumption behind the use of this dataset is that items purchased in a shopping cart are a good proxy of consumers' behavioral changes, thus allowing to capture the spread of seasonal flu reflected in a specific set of supermarket purchases. More specifically, we first identify a set of *sentinel* products whose volume of purchase is historically correlated with the previous flu season. In order to avoid the use of spurious correlations and seasonal predictors (items generally available during the flu season but not related to flu), we consider the whole purchase history of customers buying sentinel products. This allows the identification—with an Apriori algorithm—of *sentinel baskets*, i.e., products bought together that we can use as a proxy for the actual seasonal flu. By using sentinel baskets purchases, we develop a nowcasting and forecasting algorithm that provides seasonal flu incidence in Italy estimates up to 4 weeks ahead of the regular surveillance system. We make use of the Support Vector Regression (SVR) model to produce our predictions. We need to emphasize that the most important component in our framework is the data proxy—sentinel baskets—and that any other forecasting method can be applied in this framework.

Our results show that exploiting the information hidden in the retail market data can contribute to predicting the future incidence of influenza. Our findings indicate that the seasonal influenza forecast accuracy improves with the use of retail records and our predictive framework outperforms the baseline autoregressive model with historical ILI reports. More specifically, with two-week and three-week forecasts ahead, forecast performance indicators improve consistently with error estimates decreasing of about 50%. In order to support the rationale behind our choice of *sentinel baskets* as a proxy for predicting seasonal influenza, we introduce a second baseline using single products' time series of retail market data. Forecasts obtained by using *sentinel baskets* are significantly more accurate than those obtained using single products' time series. It's not the predictive power of our framework that is important, but rather the increase of the predictive power when we add the sentinel baskets that capture hidden human behaviors adapted to ongoing influenza epidemics. The presented work shows quantitatively the value of incorporating retail market data in forecasting approaches, adding one more dataset to the armory of proxy signals that can be used for the real-time analysis of epidemics. The framework developed in this paper has shed lights on the great potential of combining other predictive approaches (e.g., mechanistic models and/or deep learning models) and assimilating algorithms based on different proxy data [59], thus defining ensemble forecasting methodologies that have proven to achieve the reliability required in the policy-making process.

## Results

Our main goal is to study whether retail market data can act as a proxy for predicting influenza. Specifically, our aim is the development of influenza incidence forecasts 4 weeks in advance of the latest ground truth data released from the regular surveillance system. Generally, the release date of the ground truth is delayed by one week, according to the value of $k$, where $k$ be the $k$ week ahead ($k$ = 1, 2, 3 or 4). Therefore a distinction can be made between hindcast targets ($k$ = 1), i.e. inferring the present influenza incidence value of a week that has already passed by, nowcasting ($k$ = 2), i.e. predicting the influenza incidence value during the week in which the forecast is prepared, and forecasting ($k$ > 2), i.e. predicting the flu activity in the future weeks from the moment in time the analysis is performed.

The novelty of our work lies in the framework we design to tackle this task. We built a data-driven approach, exploiting information extracted from the retail market data, using data mining and machine learning techniques, leveraging customers' behavioral changes during the influenza peak as observed from the items they purchased in their shopping carts. We develop a nowcasting and forecasting framework that makes use of *sentinel baskets*, i.e., products bought together, to provide estimates for seasonal flu incidence in Italy up to 4 weeks ahead of the latest ground truth data.

We base our analysis on real-world data describing the purchases of the customers of COOP, one of the largest supermarket chains in Italy. This source of data has been used for different purposes, such as identifying successful innovations, meant to be a success later on [60], introducing an alternative metric to GDP by quantification of the average sophistication of satisfied needs of a population [61], creating a personal cart assistant that suggests to the customer the items to put in her shopping list based on a innovative clustering method [62] and finally, describing the buying behavior of different classes of customers, as highly ranked customers that have more sophisticated needs tend to buy niche products, i.e., low-ranked products, and on the other hand, low-ranked, low purchase volume customers tend to buy only high-ranked products, very popular products that everyone buys [63].

We generate influenza activity forecasts for the 2011/12, 2012/13, 2013/14, and 2014/15 influenza seasons. Influenza activity in Italy is officially monitored by the Italian National Institute of Health, "Istituto Superiore di Sanità" (ISS) and the Interuniversity Research Centre on Influenza (Ciri), through a system called Influnet. As ground truth data and forecast targets, we consider the ILI incidence defined as the number of patients presenting ILI symptoms over all the persons seeking medical attention during a specific week in the network of about 900 sentinel General Practitioners (GPs) and pediatricians of the Influnet system.

Fig 1 displays the predictions against the reported influenza activity level for the four time horizons, 1, 2, 3, and 4 weeks ahead. Overall predictions track the influenza activity level very accurately, as shown in the top panel of the figure. Close inspection shows that the 1 week ahead predictions from the regression model with the *sentinel baskets* and the reported influenza activity level is very similar, with small errors. For 2, 3, and 4 weeks ahead, our *sentinel baskets* continue to track rather closely the influenza activity level with some overshooting in some cases.

In order to evaluate the forecast performance of our approach we consider standard indicators such as the *Pearson correlation*, the *mean absolute percent error (MAPE)* and the *root mean square error (RMSE)* of the 1–4 weeks ahead forecast time series with respect to the ground truth provided by the Influnet system. In Table 1, we report these indicators calculated over all the influenza seasons considered here. Specifically, we report the performance of forecasts obtained by considering the top 1 and top 5 most correlated *sentinel baskets* called *Basket-1* and *Basket-5*, respectively. We test numerically that the performance remains stable,

 

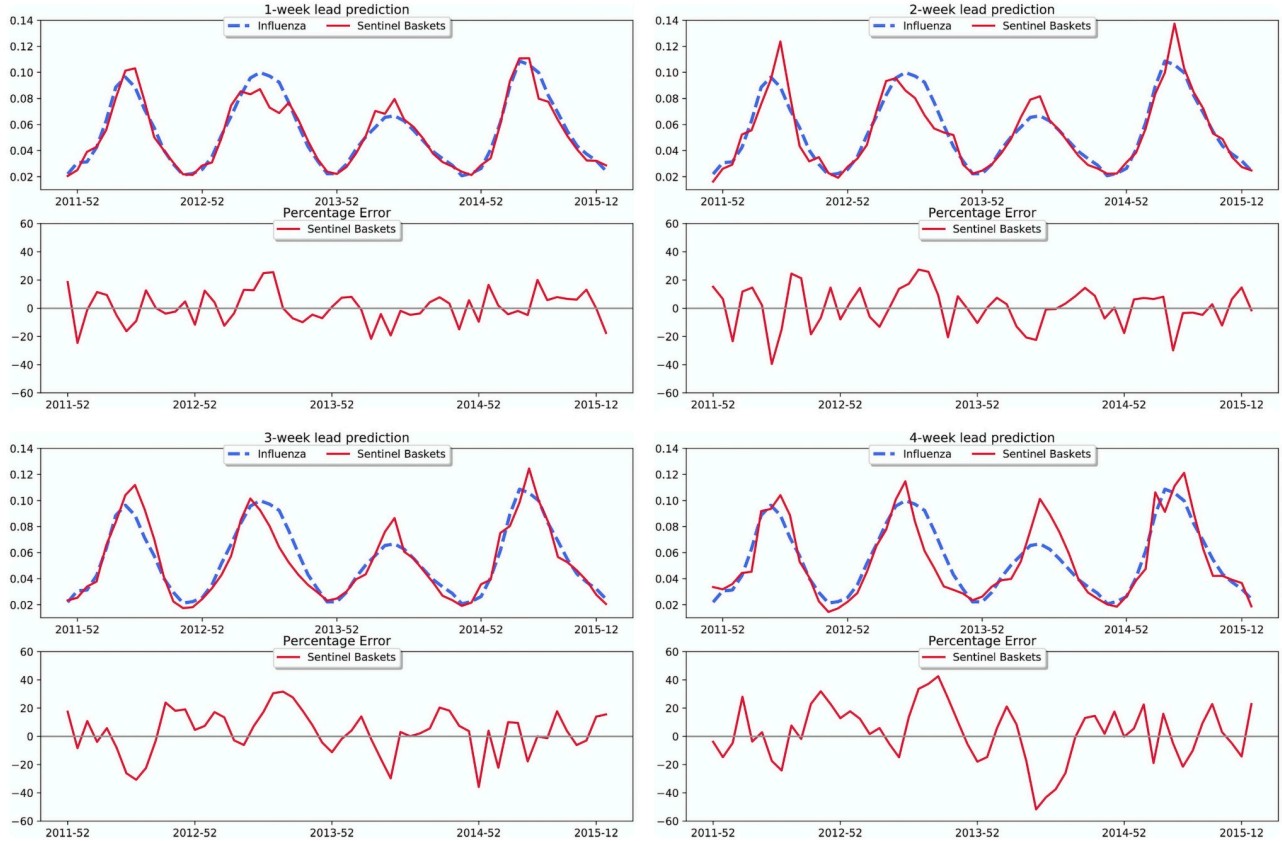

**Fig 1. Predictions for 1–4 weeks ahead.** The top plot of each panel shows the ground truth influenza activity time series along with the predictions from our framework using the *sentinel baskets*. The time dependent percentage error is displayed in the bottom plot of each panel.

increasing the number of baskets considered in the predictive framework. Along with the results from our forecast framework augmented with the sentinel basket data, we report the performance of two baseline forecast approaches: i) *autoreg*: this approach only uses historical Influnet data via the autoregressive model; ii) *Product-5*: this baseline forecast method integrates as proxy data the time series of the most correlated products put together in a basket in the same prediction model of our main approach.

From Table 1, it is evident the added value of using the *sentinel baskets* over a simple historical autoregressive approach and simple product purchases. Forecasts obtained with the

**Table 1. Performance indicators.** Performance indicators with respect to the Influnet ground truth for the sentinel basket forecast approach and the baselines (*autoreg*, *Product-5*) for the whole period 2011–2015.

| | Pearson correlation* | | | | MAPE | | | | RMSE | | | |
|---|---|---|---|---|---|---|---|---|---|---|---|---|
| | 1 week ahead | 2 week ahead | 3 week ahead | 4 week ahead | 1 week ahead | 2 week ahead | 3 week ahead | 4 week ahead | 1 week ahead | 2 week ahead | 3 week ahead | 4 week ahead |
| *autoreg* | 0.95 | 0.82 | 0.76 | 0.77 | 9.79 | 19.65 | 24.15 | 27.79 | 0.79 | 1.53 | 1.81 | 1.77 |
| *Product-5* | 0.60 | 0.49 | 0.28 | 0.01 | 41.47 | 41.80 | 44.22 | 51.07 | 2.88 | 3.07 | 3.42 | 3.76 |
| Basket-1 | **0.96** | **0.94** | **0.94** | **0.91** | **8.77** | **11.48** | **12.29** | **16.65** | **0.74** | 0.99 | **0.97** | **1.24** |
| Basket-5 | **0.96** | **0.94** | 0.93 | 0.87 | 11.80 | 13.48 | 14.77 | 17.62 | 0.75 | **0.95** | 1.02 | 1.35 |

* for all coefficients p-value < 0.01.

**Table 2. Relative efficiency.** Estimate of relative efficiency of our approach compared with the *autoreg* baseline with 95% confidence interval (CI). Relative efficiency being larger than 1 suggests increased predictive power compared with the alternative method.

| | Point Estimate | | | | 95% CI | | | |
|---|---|---|---|---|---|---|---|---|
| | 1 week ahead | 2 week ahead | 3 week ahead | 4 week ahead | 1 week ahead | 2 week ahead | 3 week ahead | 4 week ahead |
| Basket-1 vs autoreg | 1.14 | 2.36 | 3.47 | 2.05 | [0.48, 1.40] | [1.22, 3.09] | [1.15, 4.74] | [0.97, 2.52] |
| Basket-5 vs autoreg | 1.11 | 2.57 | 3.13 | 1.74 | [0.71, 1.41] | [2.00, 3.20] | [1.98, 4.14] | [0.92, 2.15] |

sentinel basket approach are significantly more accurate compared to the baseline approaches, especially in the 3 and 4 weeks ahead, time horizon. It is worth remarking that the *autoreg* baseline has a better performance in comparison to the *Product-5* baseline. As expected, we also remark that the performance of forecasts deteriorates as the time horizon increases. We report the results of individual influenza seasons 2011/12 to 2014/15 in S1 Table.

To assess the statistical significance of the improved prediction power of the sentinel basket approach, we report its relative efficiency with respect to the baseline approaches in Table 2. The relative efficiency between two approaches is defined here as the ratio of the mean-squared error of Approach 2 to that of Approach 1 [64]:

$$e(x^{(1)}, x^{(2)}) = \frac{MSE_{obs}^{(2)}}{MSE_{obs}^{(1)}} \tag{1}$$

where

$$MSE_{obs}^{(i)} = \frac{1}{n}\sum_{t=1}^{n}(x_t^{(i)} - y_t)^2. \tag{2}$$

We also report the 95% confidence interval for the relative efficiency. The relative efficiency can be estimated by the time series stationary bootstrap method [65], where the replicated time series of the error residual is generated using random blocks with mean length 14 (which corresponds to the on-season weeks with an ILI rate value greater than the threshold of 0.02).

Table 2 shows that our approach is estimated to be almost twice as efficient as the autoregressive baseline, and the improvement in accuracy is highly statistically significant. We do not include the second baseline of the single products' time series, as its predictive power proved to be rather low. Comparing our results with [59], the only influenza nowcasting and forecasting approach in Italy, where the authors used data extracted from a Web-based participatory surveillance system, we succeed in predicting influenza incidence with higher accuracy reducing the error significantly.

## Explainability—Content of sentinel baskets

An important advantage of our approach is the fact that we can explain the results. Looking into the content of our top *sentinel baskets* we can provide explanations for the predictions and comment about the nature of the products included in these baskets. Fig 2 displays the products that are present at least in one of the top 5 *sentinel baskets* sorted vertically, accordingly to the number of years that they are present.

We notice that the products vary in each season but there are several that appear in more than one. The most frequent product is oranges that appear in the *sentinel baskets* for all the seasons. Additionally, we have several fresh fruits and vegetables, such as cabbage, potatoes, pears, fennel, and mandarins. We also have prepared vegetables as well as frozen vegetables.

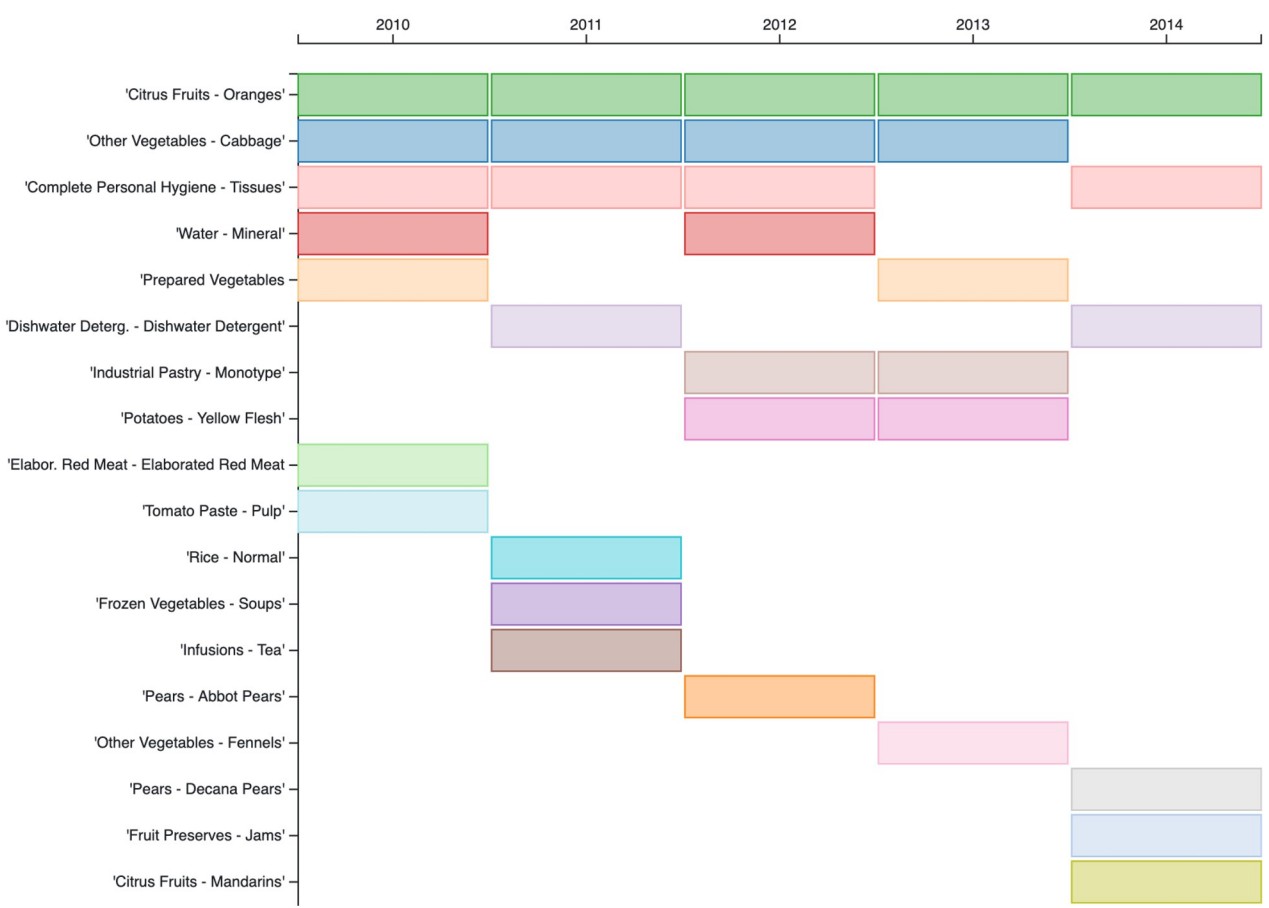

**Fig 2. Products.** Products are sorted vertically accordingly to the number of years that they are present at least in one of the top 5 baskets.

Besides fruits and vegetables, we have beverages such as mineral water and tea, and several other ingredients such as tomato paste, rice, red meat, pastry, and jam. Finally, among the most common products to appear we have tissues, under the category of personal hygiene.

The majority of these products are rather common to prepare a healthy meal or snack that could benefit a sick person. For example, one could not but expect oranges as it's a common belief that vitamin C can help strengthen the immune system. Additionally, we notice several ingredients used to prepare a vegetable soup or a broth, which are nutritious and hydrating as well as soothing when served hot.

Surprisingly, we also get dishwater detergent which is not intuitive as to its relation with influenza. Nevertheless, it is a rather common product of a supermarket basket, so it's possible to appear among the most frequent products bought together with any other "influenza" product.

## Discussion

In this study we propose the use of a novel data source, namely retail market data, as a proxy for predicting seasonal influenza. The rationale behind our choice is that customers' behavioral changes are reflected in the items purchased in a shopping basket, thus providing a valuable proxy for the spread of seasonal influenza. We make use of a regression model (SVR) to produce our forecasts for 1 to 4 weeks ahead. We need to emphasize that the most important

component in our framework is the data proxy—sentinel baskets—and that any other forecasting method can be used instead of the SVR model. We compare the results obtained with the sentinel basket approach with a baseline autoregressive model (*autoreg*) that considers only historical influenza data from the traditional surveillance Influnet.

The analysis of the results obtained for the Italian flu season from 2011 to 2015 shows the superiority of the sentinel basket approach. The forecasts consistently outperform the baseline autoregressive model, thus proving the added value of incorporating retail market data quantitatively. The retail market data we use for our approach are in the form of *sentinel baskets (Basket-1 and Basket-5)* and not just a basket of simple time series of single products, such as in the second baseline (*Product-5*), where we use the most correlated products with the influenza adoption trend. We demonstrate that the use of single products' time series does not produce the same results as using our *sentinel baskets*. The predictions of our approach are significantly more accurate than the predictions with the use of a basket of single products in all four-week forecasts. We need to stress the fact that we obtain a noticeable increase in the predictive power of our approach when we add the sentinel baskets.

Additionally, to interpret the results we examine the content of the top 5 *sentinel baskets* that are used as data proxy to predict influenza occurrence. We notice that the majority of the products make intuitive sense, as we have several fruits and vegetables, as well as ingredients for soup, tissues, water, and tea. The fact that these products are what we would expect to find among the most relevant products to influenza, demonstrates that our framework is capable to provide meaningful explanations for the final predictions.

Our retail dataset also contains OTC (over-the-counter) drugs that are available without a prescription. Unfortunately, in our study, these products rarely correlate with influenza so they don't occur among the sentinel products. A possible explanation is that when people get sick, they tend to visit the pharmacy directly and not the supermarket. Additionally, the OTC drugs are only available in the "IPERCOOP", the largest stores of the supermarket chain, and as a result, their purchase trends are not necessarily representative of the whole dataset. Finally, very often, the drugs necessary to fight influenza symptoms are everyday drugs that normally every household owns and buys independently of the influenza occurrence.

The results we present here are for influenza-like illnesses at the national level within Italy. Nevertheless, our approach shows promise to be easily extended to accurately track not only influenza in other countries where similar data sources are available but also other infectious diseases. Another important aspect is that we don't have any regional or country specific products which would make it difficult to translate to other regions or countries. Although the predictive framework is outperforming the baseline approaches, it is possible to envision the use of retail market data in the context of multi-data and ensemble approaches, thus contributing to state of the art performing forecasting schemes. Furthermore, retail data are available at the very fine geographical resolution, thus opening to the definition of proxy data for forecasting at a regional and urban level where ground truth for Influenza Incidence data are available.

## Materials and methods

In this section, we describe the data used in our study, highlighting their main characteristics. Additionally, we describe our predictive framework and its main components. We provide the data and the code of our study for reproducibility in https://github.com/jeannetm/predict_influenza_with_retail_records.

## Data description

First, we describe the influenza activity data in Italy as captured by Influnet. In addition, we describe the retail market data describing the purchases of the customers of COOP supermarkets all over Italy.

**Influenza data.**   In developed and developing countries, there are national syndromic (i.e., based on observed symptoms) surveillance systems for influenza-like illness (ILI). These systems monitor levels of ILI cases among the general population by gathering information from physicians, known as sentinel doctors, who record the number of people seeking medical attention and presenting ILI symptoms. Influenza activity in Italy is officially monitored by the Italian National Institute of Health, "Istituto Superiore di Sanità" (ISS) and the Interuniversity Research Centre on Influenza (Ciri), through a system called Influnet. The Influnet system collects data from a network of about 900 sentinel General Practitioners (GPs) and pediatricians. It compiles a weekly report in which the national and regional incidence rates by age group are published during the winter season, generally from week 42 to the last week of April of the following year (around week 17). The data cover about 2% of the Italian population. Doctors who participate in the monitoring are required to identify and write down daily, on their register, each new case of influenza. Each week, they transmit the aggregate number of cases seen by any physician (divided by age groups and by risk category) to the relevant Reference Center. The ISS processes the data at the national level and produces a weekly report. Data are published with at least one-week lag, and typically new reports provide a first estimate of the weekly ILI incidence, which is then updated in the following weeks as more data from sentinel GPs are recorded. We collected the Influnet reports for five influenza seasons, from 2011/12 to 2014/15, from week 42 to week 17. The reports are publicly available at the website of Influnet [66].

We have to mention that our analysis is performed on national influenza data because regional influenza data are not reliable enough. This is equivalent to consider that influenza spreads in a relatively homogeneous way all over the country, which for a small country as Italy is a reasonable assumption.

**Retail market data.**   We base our analysis on real-world data about customer behavior. We use a retail market dataset describing the purchases of the customers of COOP, one of the largest supermarket chains in Italy. An important dimension of the data regards the company's classification of products: there is a tree organization, and the hierarchy is built on the product typologies. The top-level of this hierarchy is called "Area" that splits the products into three fundamental categories: "Food", "No Food", and "Other" that refers to medical products. The leaves of the tree are at the bottom level of the hierarchy, called "Item". The marketing hierarchy goes like that: i) Area (3 values), ii) Macro sector (4 values), iii) Sector (13 values), iv) Department (76 values), v) Category (529 values), vi) Subcategory (2665 values), vii) Segment (7656 values), viii) Item (571092 values).

There are several conceptual issues in using the lower level of the hierarchies of the product typologies. For instance, the distinction between different packages of the same product as specified at the "Item" level, e.g., different sizes of bottles containing the same liquid, is not of interest in our study. Equally, the distinction between products of different brands, e.g., milk from company A or B, is not of interest in our study ("Segment" level). A way to solve this issue is to use the marketing hierarchy, substituting the item with its marketing "Subcategory" value. As a result, we reduce the cardinality of the dimension of the products (from 571,092 to 2,665), aggregating logically equivalent products. Throughout our study, we will refer to those subcategories as *products*.

We analyzed a dataset of 30M shopping sessions that occurred in Livorno province, one of the best-represented areas of Italy, with regards to the number of shops in the area, as well as

the number of loyal users, over 2010–2015, corresponding to about 150,000 active and recognizable customers. A customer is active if there is at least one purchase during the data time window, and she is recognizable if the purchase has been made using a loyalty card. Customers are provided with a loyalty card that allows linking different shopping sessions, and therefore reconstruct their personal shopping history. The 138 stores of the company cover the whole west coast of Italy, selling 571,092 different items. For each customer, we have N~150 baskets, D~100 different items, and an average basket length T of ~8 items.

## Predictive framework

Two main algorithmic components compose the forecasting approach proposed here: i) the *sentinel baskets* discovery from the previous influenza season $s − 1$; and ii) the use of the *sentinel baskets* for prediction in the current influenza season *s*.

The following steps summarize the definition of the *sentinel baskets* (see Algorithm 1):

- We construct the time series $S_p$ of the volume of purchases at a weekly level for each product $p \in P$. We select the *sentinel products* $\mathcal{P}$ that are more correlated with the influenza adoption trend $I$ calculating the Pearson correlation measure between $\{S_p, I\} \forall p \in P$ (see Algorithm 1, lines 2–4).

- For each of the *sentinel products* we identify the *sentinel customers*, customers $C_\mathcal{P}$ that bought them during the influenza peak $[T − 2, T + 2]$ (see Algorithm 1, line 5–9).

- For all *sentinel customers* $c \in C_\mathcal{P}$, we obtain all their purchases during the same period, and we create a pool of all their baskets $B$. We apply the Apriori algorithm to identify the most frequent baskets $B_f$. We select the baskets that are more correlated with the influenza adoption trend $I$ to be *sentinel baskets*, $\mathcal{B}$ (see Algorithm 1, lines 10–14).

---

**Algorithm 1**: Sentinel Baskets Discovery

```
  Data: Sₚ-products' time series, I-influenza time series, R-receipts
  Result: B-sentinel baskets
```
**1** $\mathcal{P} \leftarrow \emptyset; C_p \leftarrow \emptyset; B \leftarrow \emptyset; B_f \leftarrow \emptyset; \mathcal{B} \leftarrow \emptyset;$
```
// initialize the sentinel products, sentinel customers, pool of bas-
kets, frequent baskets, and sentinel baskets
```
**2 for** $p \in P$ **do**                                        // for each product
**3**    **if** $Pearson(S_p, I) > 0.2$ **then**
**4**      $\mathcal{P} \leftarrow \mathcal{P} \cup p;$                                 // add sentinel product
**5** $T \leftarrow peak(I); pi \leftarrow [T − 2, T + 2];$             // period of interest
**6 for** $rec(t, c, b) \in R$ **do**                            // for each receipt
**7**    **for** $p \in b$ **do**                          // for each product in basket
**8**      **if** $t \in pi \land p \in \mathcal{P}$ **then**
**9**         $C_p \leftarrow C_p \cup c;$                          // add sentinel customer
**10 for** $rec(t, c, b) \in R$ **do**                            // for each receipt
**11**  **if** $t \in pi \land c \in C_p$ **then**
**12**    $B \leftarrow B \cup b;$                           // add basket in pool of baskets
**13** $B_f \leftarrow Apriori(B);$              // identify the most frequent baskets
**14** $\mathcal{B} \leftarrow top5(Pearson(S_{B_f}, I));$                   // create sentinel baskets

---

Once the *sentinel baskets* have been identified, during each week of the current influenza season, $t^s$, we use their corresponding volume time series along with the past influenza incidence data in order to train a regression model of the future incidence values of influenza for 1 to 4 weeks ahead. More precisely, we proceed according to the following steps:

- We construct the composite time series, *S*, for each of the *sentinel baskets* $\mathcal{B}$, where we add the volume of purchases at a weekly level for each product $p \in \mathcal{B}$ up to week *t*.

- We introduce the regression model whose coefficients are solved by Support Vector Regression (SVR). For each forecasting week $t^s$ and forecasting target of $k$ week ahead, the SVR model is trained by the data starting from the first week in the previous season $s − 1$ to the last week $t^s − 1$.

- For the prediction, the regression model makes use of the historical ILI reports available till week $t − 1$ and *sentinel baskets* data available till week $t$.

Details on the various components of the proposed forecast framework are reported in the following sections.

**Sentinel products.**   The first necessary step to learn the *sentinel baskets* from the previous influenza season $s − 1$ is the discovery of the *sentinel products*. We need to define the time granularity of our observation period for the retail market data. We choose to use a weekly aggregation mainly because influenza reports are on a weekly base. We prepare the retail market data in order to correspond to the weekly reports of influenza, and we work on a "Subcategory" level. We report the weekly sales for each of the products $p \in P$ for all the weeks of interest (42nd week of the year until the last week of April of the following year), producing the final retail time series $S_p$.

It is crucial to notice that even working at an aggregated level in the retail hierarchy, our time series are still 2,665. So it is imperative to filter out the products that are not correlated with the influenza adoption trend $I$, so we can work mainly with products that have a similar adoption trend. We choose to use the Pearson Correlation, as it is one of the most commonly used correlation measures. In statistics, the *Pearson correlation coefficient* [67], also referred to as Pearson's $r$, is a measure of the linear correlation between two variables $x$ and $y$. It has a value between +1 and -1, where 1 is total positive linear correlation, 0 is no linear correlation, and -1 is total negative linear correlation. Using Pearson correlation coefficient we calculate the correlation $r$ between each product's time series $S_p$ with the influenza time series $I$ and we filter out the time series with a low correlation in order to identify the products that have adoption trend similar to the influenza trend, the most correlated *sentinel products* $\mathcal{P} = \{p|p \in P, r(S_p, I) > \delta\}$, where $\delta > 0.2$ to exclude products with weak or no correlation.

**Sentinel customers.**   We are interested in studying human behavior mainly during the influenza peak of the previous influenza season $s − 1$. We identify the influenza peak week at time $T$ and we define the *period of interest* $[T − \delta, T + \delta]$ where $[\delta]$ is the width of the time window. We used $[\delta] = 2$ in our experiments so we have a period of interest of 4 weeks ($\sim$1 month) which is the typical length of the period that the influenza is at its peak. Using the sentinel products in $\mathcal{P}$, we trace their sales during the period of interest, and we identify the customers that bought them through the receipts matching each customer with her corresponding purchases. These customers become our *sentinel customers* denoted with $C_{\mathcal{P}}$. We are interested in the purchases of these specific customers since those individuals would have a higher possibility to be either infected or close to an infected individual. We have to notice that customers are using loyalty cards, linking them with their purchases throughout the whole period of interest and that a loyalty card normally represents the whole household, with the probability of more than a person per household.

It is important to highlight that to produce the final predictions, we did not use these identified sentinel customers, but actually the aggregated signals. The necessity to identify the sentinel customers surges from the use of the Apriori algorithm (see subsection Sentinel baskets) that needs these detailed information to create less noisy and more robust proxy data, which are later aggregated for our predictions.

**Sentinel baskets.**   In the final step of discovering the *sentinel baskets* from the previous influenza season $s − 1$ and preparing their time series for the current influenza season $s$, we are

working backwards. Using the *sentinel customers* $C_\mathcal{P}$, we track all their purchases during the period of interest, through their receipts, and we obtain their corresponding baskets, where each basket $b$ contains products bought together under the same receipt $b = \{p_1, p_2, \ldots, p_n | p_i \in P\}$. We obtain the baskets for each customer $c \in C_\mathcal{P}$, and we create a pool of baskets $B$, discarding the information of who bought what. It is worth stressing that since we are interested in the information contained in the products bought together and in the patterns we can extract through customers behaving similarly, a key component of our approach is the *Apriori algorithm* [68].

The *Apriori algorithm* is an algorithm for frequent itemset mining and association rule learning over transactional databases. The algorithm uses a bottom-up approach where it identifies the most frequent individual items in the database and extends them to larger and larger itemsets as long as those itemsets satisfy a minimum threshold frequency. The algorithm terminates at the moment that no further successful extensions are found. It uses a breadth-first search and a Hash tree structure to count candidate itemsets efficiently. It generates candidate itemsets of length $k$ from itemsets of length $k-1$. Then it prunes the candidates who do not have a frequent sub pattern. According to the downward closure lemma, the candidate set contains all frequent $k$-length itemsets. After that, it scans the database to determine frequent itemsets among the candidates.

Using the Apriori algorithm, we extract the most frequent baskets $B_f$ in our pool. For every product in each of the most frequent baskets, we obtain the corresponding time series. Then for each basket, we create a cumulative value of all the products that belong to it, and we create the corresponding composite time series $S_{B_f}$. We use the measure presented in Step 1, and we calculate the Pearson correlation between the ILI time series $I$ and the time series for each of the baskets $S_{B_f}$, and we keep the 5-most correlated baskets, the *sentinel baskets* $\mathcal{B} \in B_f$.

In order to construct the sentinel basket time series for the current influenza season $s$, we extract the time series for each of the products that belong to the *sentinel baskets*, $p \in \mathcal{B}$ that we had obtained from the previous season. We repeat the procedure mentioned before, as for each *sentinel basket*, we create a cumulative value of all the products in it, thus creating its corresponding composite time series $S_\mathcal{B}$, $S$ for simplicity. We will incorporate these time series with the historical ILI reports in the prediction model described below.

We should also note that we learn the *sentinel baskets* only from one previous season and not more to avoid introducing biases from changes that may occur in the retail market database, as new products may appear and older disappear.

**Forecast models.**   The baseline model is inspired by Autoregression (AR) which suggests a linear relation between the current and previous values of a time series. Let $I_{t^s}$ be the logit transformed *ILI* at week $t$ in season $s$, $k$ be the $k$ week ahead ($k = 1, 2, 3$ or $4$). The baseline model could be written as

$$I_{t^s+k} = \alpha^k + \sum_{i=0}^{h-1} a_i^k I_{t^s-i}, \tag{3}$$

where $h$ is the window size, $\alpha$ and $a_i$ are the regression coefficients.

We include the *sentinel baskets'* time series $S$ as an exogenous signal, where $S_{t^s}$ be the value of $S$ at week $t$ in season $s$, yielding:

$$I_{t^s+k} = \alpha^k + \sum_{i=0}^{h-1} a_i^k I_{t^s-i} + \sum_{j=-1}^{h-1} b_j^k S_{t^s-j}. \tag{4}$$

Note that $j$ starts from $-1$ because the retail market data is up-to-date while *ILI* has one week lag such that $S_t$ has an extra week of data than $I_t$.

Further more, to test the sensitivity of the model on the number of sentinels, we expand the model as

$$I_{t^s+k} = \alpha^k + \sum_{i=0}^{h-1} a_i^k I_{t^s-i} + \sum_{n=1}^{N_S}\sum_{j=-1}^{h-1} b_j^{kn} S_{t^s-j}^n, \tag{5}$$

where $N_S$ is the total number of sentinels, and $S^n$ is the $n$th sentinel. It is essential to notice that by extending the model to incorporate more *sentinel baskets*, we can capture more shopping behaviors and with greater variance.

In the forecast model (5), $I_{t^s+k}$ is the dependent variable and $\{I_{t^s-i}\}$, $\{S_{t^s-j}^n\}$ are the the explanatory variables. The model makes use of the historical ILI reports available till week $t-1$ and sentinel baskets data available till week $t$.

Table 3 displays an example settings for the 1-week-ahead prediction at forecasting week 2015–15 with only one sentinel involved. To predict 1-week-ahead of influenza at week Apr 13, 2015–Apr 19, 2015, we use influenza data for $h$ weeks, where $h$ is the window size, until Apr 12, 2015 and sentinel data for $h + 1$ weeks until Apr 19, 2015. So for example, for $h = 5$ we use influenza data from week 2015–10 to week 2015–14 (Mar 9, 2015–Apr 12, 2015) and sentinel data from week 2015–10 to week 2015–15 (Mar 9, 2015–Apr 19, 2015). The training data starts from the start of previous season, since the *sentinel baskets* are generated from the previous season, so Oct 14, 2013 until the beginning of forecasting week Apr 12, 2015.

We make use of the Support Vector Regression (SVR) model with radial basis function (rbf) kernel in order to solve the coefficients of the above autoregression and regression models. Since the *sentinel baskets* $B_S$ for $t^s$ are generated from season $s-1$, the data earlier than that is not included in the training data. For each forecasting week $t^s$ and forecasting target $k$, the SVR model is trained by the data starting from the first week in the previous season $s-1$ to the last week $t^s-1$. For SVR model with rbf kernel, there are two hyperparameters which are regularization parameter $\mathcal{C}$ and kernel width $\gamma$ that need to be defined [69].

**Table 3. Example settings of forecast models.** An example settings for 1-week-ahead prediction at forecasting week 2015–15 with only one sentinel involved, where $h$ is the window size.

| Prediction Data | |
|---|---|
| Explanatory Variables | Dependent Variable |
| $1 \times (2h + 1)$ | $1 \times 1$ |
| ILI $1 \times h$<br>(–Apr 12, 2015) | ILI $1 \times 1$<br>(Apr 13, 2015–Apr 19, 2015) |
| Sentinel $1 \times (h + 1)$<br>(–Apr 19, 2015) | |
| Training Data | |
| Explanatory Variables | Dependent Variable |
| $52 \times (2h + 1)$ | $52 \times 1$ |
| ILI $52 \times h$<br>(Oct 14, 2013–Apr 05, 2015) | ILI $52 \times 1$<br>(Oct 14, 2013–Apr 12, 2015) |
| Sentinel $1 \times (h + 1)$<br>(Oct 14, 2013–Apr 12, 2015) | |
| 5-Fold Cross-validation<br>Hyperparameters: $h$, SVR($\mathcal{C}, \gamma$) | |

We set the range of parameters as $C \in [1, 1e4]$, $\gamma \in [0.01, 2.0]$ and window size $h \in [2, 6]$. We select the window size $h$ and SVR-related hyperparameters by Grid Search and 5-Fold Cross-validation.

**Performance indicators.** We consider the following indicators to assess the performance of the forecast approaches with respect to the ground truth influenza incidence. Our notation is as follows: $y_t$ denotes the observed value of the influenza at time $t$, $x_t$ denotes the predicted value by the model at time $t$, $\bar{y}$ denotes the mean or average of the values $y_t$ and similarly $\bar{x}$ denotes the mean or average of the values $x_t$.

*Pearson Correlation*, a measure of the linear dependence between two variables during a time period $[t_1, t_n]$, is defined as:

$$r = \frac{\sum_{t=1}^{n}(y_t - \bar{y})(x_t - \bar{x})}{\sqrt{\sum_{t=1}^{n}(y_t - \bar{y})^2}\sqrt{\sum_{t=1}^{n}(x_t - \bar{x})^2}} \tag{6}$$

*Mean Absolute Percentage Error (MAPE)*, a measure of prediction accuracy between predicted and true values, is defined as:

$$MAPE = \left(\frac{1}{n}\sum_{t=1}^{n}\left|\frac{y_t - x_t}{y_t}\right|\right) \times 100 \tag{7}$$

*Root Mean Square Error (RMSE)*, a measure of prediction accuracy that represents the square root of the second sample moment of the differences between predicted values and true values, is defined as:

$$RMSE = \sqrt{\frac{1}{n}\sum_{t=1}^{n}(x_t - y_t)^2} \tag{8}$$

In order to be similar to MAPE we multiply RMSE with 100 to make it percentage error as well.

## Supporting information

**S1 Table. Performance indicators for individual seasons.** Performance indicators with respect to the Influnet ground truth for the sentinel basket forecast approach and the baselines (*autoreg*, *Product-5*) for individual seasons 2011/12, 2012/13, 2013/14 and 2014/15. (PDF)

## Acknowledgments

We thank the supermarket chain Unicoop Tirreno and Walter Fabbri for the continuous collaboration and for providing the data that made this research possible under a rigorous privacy-preserving protocol. We also thank Daniele Fadda for his support on data visualization.

## Author Contributions

**Conceptualization:** Ioanna Miliou, Xinyue Xiong, Salvatore Rinzivillo, Qian Zhang, Giulio Rossetti, Fosca Giannotti, Dino Pedreschi, Alessandro Vespignani.

**Data curation:** Ioanna Miliou, Xinyue Xiong.

**Formal analysis:** Ioanna Miliou, Xinyue Xiong.

**Funding acquisition:** Fosca Giannotti, Dino Pedreschi, Alessandro Vespignani.

**Investigation:** Ioanna Miliou, Xinyue Xiong.

**Methodology:** Ioanna Miliou, Xinyue Xiong, Salvatore Rinzivillo, Qian Zhang, Dino Pedreschi, Alessandro Vespignani.

**Supervision:** Salvatore Rinzivillo, Qian Zhang, Giulio Rossetti, Fosca Giannotti, Dino Pedreschi, Alessandro Vespignani.

**Validation:** Ioanna Miliou, Xinyue Xiong.

**Visualization:** Ioanna Miliou, Xinyue Xiong.

**Writing – original draft:** Ioanna Miliou, Xinyue Xiong.

**Writing – review & editing:** Salvatore Rinzivillo, Qian Zhang, Giulio Rossetti, Fosca Giannotti, Dino Pedreschi, Alessandro Vespignani.

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
