## [Decision Letter · Decision Letter 0]

3 Mar 2021

Dear Dr Miliou,

Thank you very much for submitting your manuscript "Predicting seasonal influenza using supermarket retail records" for consideration at PLOS Computational Biology. As with all papers reviewed by the journal, your manuscript was reviewed by members of the editorial board and by several independent reviewers. The reviewers appreciated the attention to an important topic. Based on the reviews, we are likely to accept this manuscript for publication, providing that you modify the manuscript according to the review recommendations. In particular, both reviewers would like to have more information about the type of retail products that are predictive of flu.

Sincerely,

Cecile Viboud

Associate Editor

PLOS Computational Biology

Nina Fefferman

Deputy Editor

PLOS Computational Biology

[LINK]

Reviewer's Responses to Questions

**Comments to the Authors:**

Reviewer #1: In the current study Miliou et al present an influenza forecasting system based on combined supermarket purchases (baskets) in Italy. They compare their performance directly to autoregressive models and to autoregressive + individual item purchase performance, and indirectly to a prediction system using search queries (ref 57). They find their basket-based system systematically outperforms the other systems. This is a nice paper highlighting the use of a novel data source for pathogen surveillance. The problem I see is that it leaves me wanting more in terms of what the products are and further discussion of them. This should be straightforward for the authors to add.

Comments:

- The authors have missed a citation on NDS (of which Vespignani is a coauthor):

Althouse & Scarpino, et al. "Enhancing disease surveillance with novel data streams: challenges and opportunities." EPJ Data Science 4.1 (2015): 1-8.

(Full disclosure: I am a coauthor of this paper) A citation would be good on line 3 or 31.

- I want a lot more detail about what are in the baskets:

- What are common products?

- Are there any surprising products?

- How much do the baskets change season to season? (eg, how robust is this system?)

- Are there regional/country specific products which would make it difficult to translate to other regions/countries?

The discussion is too short and unsatisfying as is and could be a good place to go after these and other questions.

- Are the flu incidence data smoothed in figure 1?

Reviewer #2: Predicting seasonal influenza using supermarket retail records

The article study the predictive power of supermarket retail data for influenza forecast in Italy. It is delivering a relatively straight forward message that the sales record of a selected baskets (termed “sentinel baskets” by the authors) can have information towards influenza situation.

The authors emphasize the the data proxy, rather than the forecast method. The authors used Support Vector Regression (SVR) to illustrate that using supermarket retail records, they can outperform the autoregressive model with historical ILI. This is an interesting article. I think overall the message is straightforward and supported by the results. Nevertheless, I have the following comments:

1. What are the most predictive retail subcategories (i.e., your sentinel products)? Are they mostly medical products? It is important to examine whether the products make intuitive sense.

2. Why do you need to identify sentinel customers? Most big-data influenza prediction would aggregate the data over each individuals. I find it surprising that you need to identify the customers in this study. Isn’t it enough to use the aggregated purchase volume of the sentinel products? Your claim “we are interested in the purchases of these specific customers since those individuals would have a higher possibility to be either infected or close to an infected individual”, albeit sensible, seems to be tangent to this study, which is to predict ILI activity over entire population. It is also concerning from the privacy perspective.

3. How is you data correlate with other data sources? There are studies using pharmacy sales data. If one has access to pharmacy sales data, will your supermarket retail data add information on top of that? Why? The authors are welcomed to comment on this in the discussion.

**Have all data underlying the figures and results presented in the manuscript been provided?**

Reviewer #1: Yes

Reviewer #2: Yes

PLOS authors have the option to publish the peer review history of their article (what does this mean?). If published, this will include your full peer review and any attached files.

Reviewer #1: No

Reviewer #2: **Yes: **Shihao Yang

Figure Files:

Data Requirements:

Reproducibility:

References:

---

## [Editor Report · Decision Letter 1]

15 May 2021

Dear Dr Miliou,

We are pleased to inform you that your manuscript 'Predicting seasonal influenza using supermarket retail records' has been provisionally accepted for publication in PLOS Computational Biology.

Best regards,

Cecile Viboud

Associate Editor

PLOS Computational Biology

Nina Fefferman

Deputy Editor

PLOS Computational Biology

---

## [Editor Report · Acceptance letter]

1 Jul 2021

PCOMPBIOL-D-20-02083R1 

Predicting seasonal influenza using supermarket retail records

Dear Dr Miliou,

I am pleased to inform you that your manuscript has been formally accepted for publication in PLOS Computational Biology. Your manuscript is now with our production department and you will be notified of the publication date in due course.

With kind regards,

Agota Szep
